# Role of Endoscopic Ultrasound in Diagnosis of Pancreatic Ductal Adenocarcinoma

**DOI:** 10.3390/diagnostics14010078

**Published:** 2023-12-28

**Authors:** Abhirup Chatterjee, Jimil Shah

**Affiliations:** Department of Gastroenterology, Postgraduate Institute of Medical Education and Research (PGIMER), Chandigarh 160012, India; drabhirup@gmail.com

**Keywords:** PDAC, EUS, fine-needle aspiration (FNA), fine-needle biopsy (FNB), precision medicine, pancreatic carcinoma

## Abstract

Pancreatic ductal adenocarcinoma (PDAC) is the most common (90%) type of solid pancreatic neoplasm. Due to its late presentation and poor survival rate, early diagnosis and timely treatment is of utmost importance for better clinical outcomes. Endoscopic ultrasound provides high-resolution images of the pancreas and has excellent sensitivity in the diagnosis of even small (<2 cm) pancreatic lesions. Apart from imaging, it also has an advantage of tissue acquisition (EUS fine-needle aspiration, FNA; or fine-needle biopsy, FNB) for definitive diagnoses. EUS-guided tissue acquisition plays a crucial role in genomic and molecular studies, which in today’s era of personalized medicine, are likely to become important components of PDAC management. With the use of better needle designs and technical advancements, EUS has now become an indispensable tool in the management of PDAC. Lastly, artificial intelligence for the detection of pancreatic lesions and newer automated needles for tissue acquisition will obviate observer dependency in the near future, resulting in the wider dissemination and adoption of this technology for improved outcomes in patients with PDAC.

## 1. Introduction

Pancreatic ductal adenocarcinoma (PDAC) is the most common type of solid pancreatic neoplasm comprising more than 90% of all solid pancreatic neoplasms [1]. Although conventionally considered a disease of the elderly with the median age of detection being 71 years, recent data from the Surveillance Epidemiology and End Results Program (SEER) and the Center for Disease Control (CDC) have revealed an increasing rate of pancreatic cancer with a greater number of cases being detected among young individuals [2,3]. In a large population-based study from the Cedars-Sinai Medical Center group, the age-adjusted incidence rate (aIR) was found to be alarmingly increasing between 2001 and 2018 among young patients < 55 years old (more so in the group that was 15–34 years old) [2]. Further population-based cancer registry data show an increase in its annual incidence of 0.77%, 2.47%, and 4.34% in the age groups of 45–49 years, 30–34 years, and 25–29 years, respectively, and it is projected to be the second most fatal cancer by 2030 [2,3,4,5,6]. Despite its increasing incidence, pancreatic cancer remains one of the most fatal malignancies with a grave 5-year survival rate of only 10–12% [2]. The major hindrance in improving patient prognosis is the lack of clearly defined risk factors, lack of targeted surveillance protocols or populations, and absence of specific symptoms or specific biomarkers. Due to these factors, the clinical presentation of patients is delayed, and only 10–15% of patients are in an operable stage at the time of presentation [3,7]. So, early diagnosis and treatment are of utmost importance in improving the prognosis of PDAC. Since its introduction, endoscopic ultrasound (EUS) has been used for the detection of pancreatic mass lesions and their characterization. Initially, only B-mode imaging was available in EUS for the characterization of lesions. However, with advancements in technology, various imaging modes like EUS elastography and contrast-enhanced EUS (CE EUS) are now available for better lesion characterization. Similarly, since its introduction, EUS-guided fine-needle aspiration (FNA) has become a cornerstone in tissue diagnoses of pancreatic mass lesions. Various dedicated aspiration and biopsy needles have been developed over the last two decades, providing a high diagnostic accuracy and high-quality histological core to perform molecular studies. Apart from its diagnostic role, EUS has a therapeutic implication in the management of various aliments associated with pancreatic carcinoma, like celiac plexus neurolysis or RFA (radiofrequency ablation) for pain management, EUS-guided biliary drainage (as a primary modality or rescue therapy), EUS-guided gastrojejunostomy for the management of gastric outlet obstructions, and EUS-guided RFA for locally advanced tumors [8,9]. In this dedicated review, we focus our discussion on the recent advances and the current role of EUS in the diagnosis of PDAC.

## 2. EUS in the Detection of Tumors

EUS is now a well-established modality in the detection of pancreatic lesions. Due to its proximity with the entire pancreatic parenchyma from the stomach to duodenum and good spatial resolution, even small lesions can be detected with a high accuracy. Compared to abdominal ultrasound, in which the entire pancreas might not be evaluated due to gas artifacts or body habitus, EUS has a better accuracy in detecting pancreatic lesions by obviating both these problems. Studies have shown that the sensitivity of EUS is higher than that of transabdominal ultrasound and computed tomography (CT) scans (94% vs. 67% vs. 74%; *p* < 0.05) in the diagnosis of PDAC [10]. EUS is even more clinically useful in the presence of small lesions (<2 cm), as it provides a higher detection accuracy compared to that of CT or magnetic resonance imaging (MRI). In a study by Muller et al., EUS had a better sensitivity compared to that of CT and MRI for the detection of small pancreatic lesions < 30 mm (93% vs. 53% and 67%, respectively) [11]. Similarly, in another study by Sakamoto et al., EUS proved to be superior to CT for lesions smaller than 20 mm with a sensitivity of 94.4% (vs. 50%) [12]. Additionally, for lesions smaller than 10 mm, EUS has shown a higher sensitivity compared to that of transabdominal ultrasound, CT, and PET scans (sensitivity > 80%, 17–70%, 33–75%, and 50%, respectively) [13]. In a meta-analysis involving 206 patients with suspected pancreatic masses but indeterminate CT scans, EUS showed a pulled sensitivity, specificity, and accuracy of 85%, 58%, and 75%, respectively, in the detection of pancreatic lesions. Pancreatic masses were diagnosed in 70% of patients (42% were adenocarcinoma) with a mean tumor size of 21 ± 1.2 mm, which highlights the impact of EUS for small lesions of the pancreas [14]. Similarly, EUS is more accurate compared to MRI in the detection of solid pancreatic lesions (100% vs. 22%; *p* < 0.001) in high-risk individuals during the surveillance period [15].

For B-mode EUS imaging, PDAC usually appears as a heterogenous, hypoechoic mass lesion with irregular borders. There might be upstream parenchymal atrophy with or without main pancreatic duct dilation [16]. Pancreatic in situ carcinoma might present as just focal narrowing/stricture in the pancreatic duct with surrounding hypoechoic areas [17]. However, some lesions can also have atypical features, such as the presence of a small amount of calcification, cystic areas, or isoechoic lesions, compared to surrounding pancreatic parenchyma. Compared to this finding, pancreatic neuroendocrine tumors usually present as well-circumscribed, homogenous, hypoechoic lesions with clearly regular borders. In comparison, autoimmune pancreatitis might have a homogenous lesion that is hypoechoic in nature with parenchymal heterogenicity and bile duct wall thickening [16]. However, these findings are not entirely specific to PDAC; other lesions like mass-forming chronic pancreatitis, lymphoma, pancreatic tuberculosis, or metastatic tumors in the pancreas can also produce similar findings. So, histological prediction is difficult in some patients when it is only based on B-mode imaging, necessitating the need for tissue acquisition from the same setting.

## 3. Role of EUS in the Staging of Tumors

EUS is helpful in the locoregional staging of pancreatic adenocarcinoma in terms of vascular involvement, lymph node metastases, the detection of small hepatic metastasie, and the diagnosis of minimal ascites or peritoneal carcinomatosis. Studies have shown that EUS is more accurate in the detection of vascular involvement compared to CT scans, especially in the detection of venous involvement. The sensitivity and specificity of EUS for the detection of tumor vascular invasion range from 42% to 91% and from 89% to 100%, respectively, in different studies [10]. In a meta-analysis involving 30 studies with 1554 patients, the pooled sensitivities of EUS and CT were 72% and 63%, respectively, and the pooled specificities of EUS and CT were 89% and 92%, respectively. In a sub-group analysis of nine studies in which EUS and CT were both performed, CT showed a lower sensitivity compared to that of EUS (48% vs. 69%) [18]. Moreover, the sensitivity of EUS is also higher for the detection of portal vein involvement compared to superior mesenteric vein/artery or celiac artery involvement. This may be due to technical difficulties in providing entire images of these vessels, due to obscuration by large tumors in the uncinate or inferior portion of the pancreatic head [10]. Contrast-enhanced EUS (CE EUS) can be also of value in detecting subtle portal venous involvement missed by conventional EUS or CT [19,20]. There are four types of vascular involvement in pancreatic cancer: type 1, clear invasion with the encasement of a vessel by a tumor; type 2, a tumor that is in contact with a vessel with the loss of the hyperechoic vessel layer with or without vessel irregularity or luminal narrowing; type 3, a tumor that contacts a vessel without the loss of the hyperechoic vessel layer; and type 4, clear non-invasion with distance between the tumor and vessel [21]. Apart from that, tumor thrombus within the vessel or the presence of collaterals surrounding the tumor can also be found [22] (Figure 1).

For N staging, a meta-analysis has also shown that EUS has a better sensitivity (58% vs. 24%) and a similar specificity (85% vs. 88%) to those of CT in detecting lymph node involvement [23,24]. Malignant lymph nodes on EUS usually have a size > 10 mm, a round shape, a sharply demarcated border, and are hypoechoic in nature [25].

EUS also has a beneficial role in the diagnosis of small hepatic malignancies. In a prospective study by Okasha HH et al., EUS could detect 7.9% of incremental focal liver lesions and 5.8% of liver metastases that were missed by CT and MRI imaging, with the median size of the hepatic lesions being 12 mm [26]. From another prospective study by Singh P et al., EUS also showed a better sensitivity compared to that of CT in the diagnosis of small liver lesions (40% vs. 19%; *p* = 0.008) [27]. EUS has a better accuracy in detecting minimal ascites or omental thickening in pancreatic carcinoma, and sampling can be performed withing the same session for the detection of metastases to avoid futile surgeries [28,29]. Similarly, a diagnosis of perivascular cuffing suggestive of extravascular migratory metastases (EVMMs) can be made via EUS. In a study by Rustogi T et al., EUS could detect EVMMs that were initially missed by CT or MRI as perivascular cuffing in an additional 28% of patients with PDAC. In that study, the disease was upstaged in 14 patients from resectable to unresectable after EUS-guided FNA of EVMMs [30]. Similarly, in a recent meta-analysis involving 795 patients, EUS could identify unresectable disease in 14% of patients in whom initial cross-sectional imaging showed resectable disease [31] (Figure 2). So, EUS is an extremely valuable modality in patients with potentially resectable disease for accurate T and N staging as well as the diagnosis of previously missed hepatic metastases, ascites, or EVMMs, so that futile laparotomies can be avoided. However, being operator-dependent, interobserver variability and the availability of a trained physician are of utmost importance in providing optimal clinical results. The National Comprehensive Cancer Network (NCCN) and European Society of Medical Oncology (ESMO) guidelines also recommend EUS for equivocal pancreatic lesions that are iso-dense in CT and the assessment of venous involvement. A biopsy may be obviated in resectable cases of patients who are scheduled for direct surgery; however, tissue diagnosis is required for borderline or locally advanced lesions before starting chemotherapy [32,33].

## 4. Role of Contrast-Enhanced EUS

CE EUS allows for the characterization, differential diagnosis, and accurate staging of pancreatic lesions. Typically, pancreatic carcinoma is hypo-enhancing, neuroendocrine tumors are hyper-enhancing, and autoimmune pancreatitis/mass-forming pancreatitis is iso-enhancing compared to surrounding pancreatic parenchyma on CE EUS [34] (Figure 3). A recent meta-analysis has shown a pooled sensitivity and specificity of 91% and 86%, respectively, in the diagnosis of pancreatic adenocarcinoma [35]. Apart from the qualitative image, a quantitative analysis using a TIC (time-intensity curve) can be also of value in evaluating CE EUS images. A TIC shows peak enhancement values, which help in the differentiation of chronic pancreatitis from pancreatic carcinoma [36]. Moreover, the peak enhancement value also shows a correlation with microvascular density in histological analyses [37]. CE EUS has also been shown to perform better compared to conventional EUS in characterizing vascular invasion by mass lesions. After contrast injection, portal vein borders can be delineated better, resulting in a higher accuracy compared to that of conventional EUS or CT scans. In a single-center, retrospective study by Nakai A et al., the diagnostic accuracy of CE EUS was better than that of B-mode EUS or CT for diagnosing portal vein invasion (93.2%, 72.7%, and 81.8%, respectively; *p* < 0.05) [19]. In a similar study by Imazu H et al., the accuracy of the T staging of CE EUS was better compared to that of conventional EUS (92.4% vs. 69.2%; *p* < 0.05) [38]. Similarly, prospective studies have also shown that the use of contrast-enhanced EUS (CE EUS) has a better sensitivity compared to that of CT or EUS in detecting small liver lesions < 10 mm in size (93.3% vs. 84.4% vs. 85.6%, respectively; *p* < 0.05) [39,40]. On CE EUS, lymph nodes can also be examined for enhancement patterns, with malignant lymph nodes showing a heterogeneous enhancement pattern compared to benign lymph nodes. In a study by Miyata T et al., the heterogenous enhancement pattern on CE EUS had better sensitivity compared to conventional EUS for the assessment of lymph nodal metastases [41]. Apart from its diagnostic role, CE EUS also helps in the selection of the target area for EUS-guided FNA. In a recent meta-analysis involving six studies and 701 patients, the pooled sensitivity of CE EUS-guided FNA was higher than that of EUS-guided FNA (84.6% vs. 75.3%; *p* < 0.001) [42]. However, in a recent randomized trial comparing CE EUS-guided FNB and conventional FNB with fanning techniques, both arms had a similar diagnostic accuracy with similar requirements for the median number of passes [43]. So, with the advent of newer EUS FNB (fine-needle biopsy) needles, the routine use of CE EUS before EUS tissue acquisition (EUS TA) remains questionable. 

## 5. EUS Elastography

Like CE EUS, EUS elastography has also been used for the better characterization of pancreatic mass lesions. Two types of EUS elastography are strain elastography and shear wave elastography. In strain elastography, interpretations can be both qualitative (using a color pattern) or quantitative (using the strain ratio (SR) or strain histogram (SH)). On qualitative examination, pancreatic malignant tumors are heterogeneous, predominantly having a blue pattern (hard signal) with small green areas and a geographic appearance compared to pancreatic neuroendocrine malignant lesions, which have a homogenous blue pattern [44] (Figure 4). In a quantitative analysis, a SR greater than 10 represents hard tissue (likely to be malignant) and <10 is intermediate (likely to be benign) [45]. A SH mean < 50 indicates hard tissue suggestive of a malignancy, whereas a SH of 50–150 (intermediate) and >150 (intermediate-soft) indicate it is more likely to be benign [45]. In a study among 86 consecutive patients with solid pancreatic lesions, the mean SR for adenocarcinoma was 18.12 (95% CI 16.03–20.21) and for inflammatory masses was 3.28 (95% CI 2.61–3.96), and both were significantly higher compared to those of a normal pancreas (mean SR of 1.68) [45]. In a subsequent study, authors showed that for malignant pancreatic tumors, at a cut-off of a SR >10 and a SH < 50, the sensitivity and specificity were 100% and 92.3%, respectively [46]. In another study, the use of EUS elastography was superior to dynamic CT and B-mode EUS with a higher sensitivity, specificity, and accuracy for staging pancreatic carcinoma with a better delineation of vascular involvement [47]. Facciorusso A et al. evaluated the utility of EUS-elastography-guided FNA for pancreatic lesions, and its diagnostic accuracy, sensitivity, and specificity were favorable (94.4%, 93.4%, and 100%, respectively) [48]. As it is a simple inbuilt procedure within the existing EUS system, no extra cost is involved for the patient. In recent times, shear wave elastography (SWE) has also been introduced into the EUS system; however, in the absence of prospective data and standardization, its role in pancreatic carcinoma is yet to be explored [49].

## 6. EUS-Guided Tissue Acquisition: Techniques and Variations

In recent years, the role of neoadjuvant chemotherapy has significantly expanded to improve clinical outcomes in patients with borderline resectable and locally advanced PDAC. Moreover, recently, neoadjuvant chemotherapy has also been explored in resectable PDAC to improve clinical outcomes. Subsequently, the role of pre-operative tissue diagnosis for neo-adjuvant chemotherapy and precision medicine has also expanded in recent times [32,50]. EUS tissue acquisition is a procedure in which tissue is procured from the target lesion under endosonographic guidance using dedicated needles to establish a tissue diagnosis. Apart from tissue acquisition from the primary tumor, EUS also offers a sampling of locoregional and distant lymph nodes, concomitant liver lesions (suspicious metastases), and minimal ascites, which have a seminal role in management decisions. Studies have shown that EUS TA has a better accuracy compared to that of EUS- or CT-guided tissue acquisition in patients with pancreatic lesions [51]. Moreover, in the last decade, EUS TA has undergone a paradigm shift due to the availability of different needles, different modes of suction, and sample handling techniques to yield a better tissue material equivalent to the histological core [52].

### 6.1. EUS FNB and FNB Needles

#### 6.1.1. EUS FNB Needles

Currently, a plethora of EUS FNB needles are available from various manufacturers (for example, EZ Shot 3 Plus, Olympus; Echotip Ultra, Cook; Expect and Expect Slimline, Boston Scientific) in different diameters, starting from the stiffer 19G needles to the more flexible 20G, 22G, and 25G needles. Stiff 19G needles, despite their presumed superior diagnostic accuracy, are difficult to maneuver, especially for head lesions, and can lead to scope trauma and technical failures [53]. To obviate this problem, 19G Nitinol needles were designed with better flexibility. Laquiere A et al. conducted a randomized trial comparing the 19G nitinol needle with the 22G needle for a trans-duodenal puncture from pancreatic solid lesions. The 19G needle had a lower technical success rate (86.4% vs. 100%; *p* = 0.003), lower diagnostic accuracy (69.5% vs. 87.3%; *p* = 0.02), and inferior ergonomic score (*p* < 0.001) compared to the 22G needle [54]. The 25G needle is more flexible, easily maneuverable, has a lower risk of blood contamination and improved accessibility for smaller and hypervascular lesions, and is preferred for sampling from difficult-to-access sites like uncinate process lesions and from more fibrous solid lesions. Madhoun MF et al. conducted a meta-analysis of eight studies involving 1292 subjects and the 25G needle had a higher sensitivity compared to that of the 22G (93% vs. 85% respectively) with a similar specificity (97% vs. 100%) [55]. However, two other meta-analyses have failed to show any significant difference between the 22G and 25G EUS FNB needles [56,57]. Recent ESGE Guidelines on EUS-guided tissue acquisition also recommend 22G or 25G needles for EUS TA from pancreatic solid lesions [58].

#### 6.1.2. EUS FNB Needles

Though EUS FNA needles can provide tissue for cytological yield, their diagnostic accuracy is limited in the presence of pauci-cellular tumors or marked desmoplastic reactions as well as in situations in which the examination of tissue stroma is essential, like in lymphoma. EUS FNB has the advantage over FNA due to its ability to acquire a higher cell count, maintain tissue architecture, and provide tissue for molecular profiling [59]. EUS FNB needles have evolved significantly over the years in terms of their design and technical modifications. EUS FNB needles can be divided into first-generation true-cut needles (Quick core, 19G, Cook Endoscopy, Limerick, Ireland), second-generation reverse-bevel needles (Echo Tip Procore needle (19, 22, and 25G) Cook Endoscopy, Limerick, Ireland), and third-generation needles (fork-tip needle, e.g., shark core needle by Medtronic, Minneapolis, MN, USA; Franseen tip needle, e.g., Acquire needle by Boston Scientific, Marlborough, MA, United States; SonoTip TopGain needle by Medi-Globe, Achenmühle, Germany; and the forward-facing bevel needle with core-trap technology, e.g., Procore needle by Cook Medical, Limerick, Ireland) (Table 1). Though earlier studies on first-generation needles showed equivalent diagnostic yields between EUS FNB and EUS FNB (although fewer passes were required in the former group), subsequent high-quality prospective studies and randomized controlled trials using second- and third-generation EUS FNB needles show the superiority of EUS FNB over FNA [60]. In a multi-center RCT by Cheng et al., a subgroup analysis for patients with pancreatic masses showed the superior diagnostic accuracy of the reverse-bevel EUS FNB needle over the 25G EUS FNB needle (93% vs. 82%, respectively, *p* < 0.01) [61]. In a similar study by van Riet et al., the forward-bevel Procore 20G EUS FNB needle a showed superior diagnostic accuracy (87% vs. 78%, *p* = 0.002) and histological tissue yield (77% vs. 44%, *p* < 0.001) compared to those when using EUS FNB needles [62]. In a recent meta-analysis by Renelus BD et al. involving 19 studies (1365 patients), FNB had a better diagnostic accuracy (87% vs. 81%; *p* = 0.005), better cytopathological accuracy (87% vs. 81%; *p* = 0.005), and reduced number of passes required (1.6 vs. 2.3; *p* < 0.001) with similar adverse event rates (1.8% vs. 2.3%; *p* = 0.64) compared to FNA [63].

Regarding the various types of FNB needles, an RCT compared the 22G Franseen needle (Acquire) with the 20G forward-bevel Procore needle, in which the Franseen needle showed a significantly higher tissue length (mean length of 11.4 mm vs. 5.4 mm, *p* < 0.001) and surface area (mean surface area of 3.5 mm^2^ vs. 1.8 mm^2^, *p* < 0.001) of the sample compared to those of the 20G Procore needle. Higher diagnostic adequacy was obtained with the Franseen needle (87% vs. 67%, *p* = 0.02) [64]. In other studies, in suspected type 1 autoimmune pancreatitis patients, similar results were shown [65]. In randomized trials, Franseen needles and fork-tip needles were similar in terms of diagnostic adequacy (94.9–96% vs. 92–97.2%), and accuracy (92.3% vs. 94.4%) [66,67]. In a systematic review and meta-analysis among patients with solid mass lesions, both fork-tip and Franseen needles had similar diagnostic yields (92.8% vs. 92.7%, *p* = 0.98), with and without using ROSE (95.9 vs. 93.7%; *p* = 0.25) and between ≤two and >two needle passes (90.6% vs. 93.3%; *p* = 0.56) [68]. In a recent network meta-analysis of 16 RCTs (*n* = 1934) comparing different types of FNB needles in sampling pancreatic solid masses, Franseen and fork-tip needles were the two best-performing EUS FNB needles with significant advantages over reverse-bevel needles and EUS FNB needles for diagnostic accuracy and tissue adequacy. However, no significant difference was found between Franseen and fork-tip needles. Regarding diagnostic accuracy, both 22G Franseen and fork-tip needles were superior to 22G reverse-bevel needles (RR, 1.22 (95% CI, 1.03–1.44) and 1.19 (95% CI, 1.03–1.38), respectively). For sample adequacy, the 25G Franseen needle was superior to the 22G reverse-bevel needle (RR, 1.12 (95%CI, 1.02–1.22)). The 22G Franseen needle was found to be the best-performing EUS FNB needle regarding diagnostic accuracy, and the 22G and 25G Franseen needles followed by the 22G fork-tip needle were the two best-performing needles regarding sample adequacy [69]. Moreover, apart from diagnostic accuracy, these needles have also shown better performance in terms of the genetic profiling of tumors, the detection of actionable molecular alterations, and microsatellite instability (MSI), which have significant implications in personalized treatment as discussed below.

### 6.2. Technical Aspects in EUS TA

To maintain tissue integrity, improve the cellularity of the sample, and reduce blood contamination, various suction and sampling techniques have been described in the literature, including dry suction, wet suction, capillary suction with the stylet slow-pull technique (CSSS), door knock, and fanning techniques. In the conventional suction method, 10 mL of negative pressure is usually used, though a higher negative pressure has shown superior tissue adequacy in studies [70,71,72]. The wet suction technique was found to be superior in terms of tissue adequacy and cellularity in prospective randomized studies [73,74,75]. In a systematic review and meta-analysis of six studies (*n* = 418) including three RCTs, the wet suction technique was shown to be superior to the dry suction technique in terms of sample adequacy (pooled OR of 3.18; 95% confidence interval (CI): 1.82–5.54; *p* = 0.001) with no significant difference in blood contamination and histological diagnosis [76]. The stylet slow-pull technique has shown comparable results to conventional suction techniques with a mean of two passes for solid pancreatic lesions [77,78,79]. For cirrhotic patients and in the presence of coagulopathy, suction is usually avoided as it increases contamination in blood [80]. Keeping the stylet provides a few additional advantages by increasing the stiffness of the needle and helping remove material obtained from gastrointestinal wall punctures [80]. In a recent network meta-analysis of 16 studies (*n* = 2048), these techniques were compared (including no suctioning and the wet suction, dry suction, and stylet slow-pull techniques), and no difference among the various suction techniques was found in terms of their tissue adequacy, diagnostic accuracy, and moderate-to-high levels of cellularity. When adjusted for the effect of the type of needle used, no difference was found among these techniques in terms of the bloodiness of the sample [81]. So, the decision to use suctioning and the type of suctioning should be made using more of an individualized approach, depending on the type of lesion, presence of comorbidities, and operator experience.

Regarding the different techniques of EUS TA, the fanning technique and door-knock technique are the most used techniques during the needle pass. In the fanning technique, the elevator and big wheel of the scope are used to change the trajectory of the needle to improve diagnostic accuracy by sampling from different areas of the lesion. In a randomized trial by Bang JY et al., the fanning technique was found to require a fewer number of passes for diagnosis (median of 1 (interquartile range 1–3) vs. 1 (1–1); *p* = 0.02) and had higher accuracy of diagnosis after the first pass (57.7% vs. 85.7%; *p* = 0.02) compared to the standard technique [82]. In a prospective study, the slow-pull plus fanning technique has been shown to have a superior diagnostic accuracy (88% vs. 71%, *p* = 0.044) and less blood contamination (77% vs. 56%, *p* = 0.041) compared to the standard suction technique [83]. In the door-knock technique, the needle hits the handle after a sharp smart movement into the target, which produces a knocking sound. In a multicenter prospective cross-over trial, the tissue acquisition rate of the door-knock technique was found to be similar to that of the conventional method; however, the tissue acquired from the door-knock technique had high levels of cellularity compared to those acquired from the conventional method (54.9% vs. 41.5%, *p* = 0.03) [84].

Regarding the required optimum number of passes and actuation during EUS TA, Uehara et al. conducted a retrospective study in which they showed the optimum number of passes required for lesions in the head of the pancreas with a size less than 15 mm was three. For a lesion in the head region that is >15 mm in size or a lesion in the body/tail that is < 15 mm in size, the optimum number of passes was two, and for a lesion in the body or tail region >15 mm in size, only one pass was adequate. The overall sensitivity was 93% using this approach [85]. Recently Paik WH et al., did a randomized trial comparing various suction methods and actuation per pass for pancreatic solid mass. In that study, 15 actuation per pass had better diagnostic accuracy compared to 10 actuations with non-suction techniques. However, when using suction or CSSS, diagnostic accuracy between 10 and 15 actuation was similar [86]. However, both these studies were performed using FNA needles. Zhou W et al. conducted a randomized trial using the 22G FNB needle for the sampling of pancreatic solid lesions. The authors recommended at least three passes using suctioning or four passes without suctioning for the optimal diagnostic accuracy in patients with solid pancreatic lesions [87]. Takahashi K et al. performed a multicentric randomized trial comparing three vs. twelve to-and-fro (TAF) movements in patients with solid pancreatic tumors using the FNB needle. In that study, the diagnostic accuracy of three TAFs was not inferior to that of twelve TAFs (88.6% vs 89.5%) with less blood contamination with three TAFs compared to twelve TAFs [88].

### 6.3. Role of On-Site Evaluation of the Sample (ROSE and MOSE)

#### 6.3.1. ROSE in EUS TA

The presence of on-site cytopathologists increases the likelihood of adequate tissue sampling and helps in real-time diagnoses to deter endosonologists from taking unnecessary extra samples. However, they have limited availability, especially in resource-constrained settings with an added cost that might not be justifiable in routine clinical practice. In an initial study by Wani et al., the addition of ROSE (rapid on-site evaluation) was associated with fewer passes (the median number of passes was four vs. seven, *p* < 0.0001) with a similar diagnostic accuracy, procedure time, number of adverse events, and cost when compared to those of EUS FNB without a cytopathologist [89]. Subsequently, in a randomized non-inferiority trial, seven passes without an on-site pathologist were shown to be non-inferior to EUS FNB with an on-site pathologist (the absolute difference was 0.2%) [90]. Recently, Crino SF et al. conducted a multi-center randomized trial comparing EUS FNB with or without ROSE in patients with solid pancreatic lesions. Both arms showed a similar diagnostic accuracy (96.4% vs. 97.4%; *p* = 0.396). EUS FNB without ROSE had a higher tissue core rate (70.7% vs. 78.0%; *p* = 0.021) and a reduced procedure time compared to EUS FNB with ROSE [91]. Similarly, Chen YI conducted a multicentric trial comparing EUS FNB with EUS FNB+ROSE in patients with pancreatic lesions. Both arms had a similar diagnostic accuracy (92.2% vs. 93.3%; *p* = 0.72) with a lower number of passes required in the EUS FNB group (2.3 vs. 3.0; *p* < 0.001) compared to the EUS FNB + ROSE group [92]. These studies show that the routine use of ROSE in the presence of newer generations of FNB needles is limited.

#### 6.3.2. MOSE in EUS TA

Macroscopic on-site evaluation (MOSE) was first introduced by Iwashita et al., who described the macroscopic visible core (MVC) as whitish or yellowish bulky tissue fragments in the absence of liquid and paste-like material. In that study, the investigators used the 19G EUS FNB needle and found that an MVC > 4mm indicates sample adequacy and can improve diagnostic adequacy. In their multivariate analysis, pancreatic lesions (OR 2.92, 95% CI: 1.06–9.09, *p* = 0.03) and an MVC ≤ 4mm (OR 15.12, 95% CI: 5.81–45.02, *p* < 0.0001) had higher false negative results [93] (Figure 5). In a prospective study among 204 patients using the 22G Franseen tip EUS FNB needle for sampling solid pancreatic lesions, the authors found a high accuracy at an MVC cut-off length of ≥3 mm, especially when scheduled for next-generation sequencing (NGS) [94]. In a multicentric randomized trial, involving lesions more than 2 cm in size, EUS TA with MOSE had a similar diagnostic accuracy (92.6% vs. 89.3%, *p* = 0.37) to that of conventional EUS TA with fewer passes required (two vs. three; *p* < 0.001) [95] (Table 2).

### 6.4. Role of Repeat EUS TA

Although EUS TA (especially EUS FNB) has shown excellent sensitivity and diagnostic accuracy, studies have shown that 5–10% of cases might have inconclusive results for which the role of repeat EUS TA has been explored. In retrospective studies, repeat EUS FNB in solid pancreatic lesions has been shown to reveal diagnoses in three-fourths of cases with second EUS FNB showing a sensitivity of 80% [119]. In another study, the location of the lesion, number of needle passes, type of needle used, diameter of the needle, and use of suction were found to influence the performance of a repeat EUS TA. In this study, body or tail lesions (vs. head, *p* = 0.005), ≥four passes (vs. ≤three passes, *p* = 0.011), the FNB needle (vs. FNA needle; *p* = 0.004), the 22G needle (vs. 19/20G needle; *p* = 0.014), and the use of suctioning (vs. other methods; *p* = 0.020) have been shown to improve the diagnostic yield in solid pancreatic lesions in a multivariate analysis [120].

### 6.5. Effect of Biliary Drainage on EUS TA

There is no clear consensus to date regarding the appropriate chronology of EUS TA and ERCP for extrahepatic biliary obstructions [121,122]. EUS TA has its limitations when there is a biliary stent in situ (plastic or metal). This is due to stent-related artefacts, stent-induced local inflammation, acoustic reverberations, and shadowing [123,124]. Studies have reported there is a risk of understaging periampullary lesions resulting in unjustified laparotomies due to the presence of biliary stents [125]. In a recent meta-analysis of nine studies, EUS TA was shown to have lower diagnostic accuracy (for reportedly confirmed malignancies; OR of 0.58; 95% CI, 0.46–0.74; *I*2 = 0.0%) and comparable tissue inadequacy (OR of 1.12; 95% CI, 0.76–1.65; *I*2 = 0.0%) in groups in the presence of a stent compared the absence of a stent [115].

### 6.6. Complications of EUS TA and Risk of Needle Tract Seeding (NTS)

EUS TA is minimally invasive and the risk of complications is minimal, which includes abdominal pain, pancreatitis, bleeding, perforation, infection, and needle track seeding (NTS) with an overall frequency ranging from 0 to 3% [80,126]. The majority of complications are mild and can be managed conservatively. NTS is an anticipated risk of EUS TA especially for body and tail lesions of the pancreas when sampled through the trans-gastric route. Park J S et al. conducted a retrospective study of 528 patients with distal pancreatic cancer who underwent distal pancreatectomies. Among these, 193 patients had undergone EUS FNB before surgery, and 335 had not. After resectioning, the recurrence rates were comparable amongst both groups (EUS FNB: 72.7%; non-EUS FNB: 75%; *p* = 0.58) at a median follow-up of 21.7 months and there was an equal cancer-free survival rate across both the groups (*p* = 0.58) [127]. In a retrospective study by Yane et al., the 5-year cumulative risk of NTS was shown to be 3.8% (95% CI 1.6–7.8%) without significantly affecting overall survival and median recurrence-free survival rates [128]. In a recent meta-analysis involving 10 studies (13,238 patients), the pooled rate of NTS was 0.3%. There was no difference in terms of metachronous peritoneal dissemination observed between patients who underwent EUS TA and non-sampled patients [42]. So, NTS after EUS TA remains an important parameter to consider before performing EUS TA; however, it is not associated with reduced cancer-free or overall survival.

In the authors’ personal experience, EUS TA starts initially with an optimum B-mode image examination, the localization of masses and their vascular involvement, and the confirmation of the presence or absence of local lymphadenopathy. For EUS TA, a 22G Franseen needle is most commonly used, with the 25G needle reserved for lesions at difficult locations like in the uncinate process. All attempts are made to perform EUS TA from the duodenal station rather than from the gastric station to avoid needle track seeding. During the needle pass, the fanning technique and wet-suction technique are routinely used. Three passes with a macroscopic examination of the sampled tissue for material adequacy are routinely performed. ROSE is usually reserved in case of previous inconclusive results during a repeat EUS TA.

## 7. Recent Advancement in EUS TA in Pancreatic Carcinoma

### 7.1. Role of EUS TA in the Era of Precision Medicine

In oncology, precision medicine comprises diagnoses as well as targeted therapies based on genetic profiles, host factors, and environmental factors. In PDAC, precision medicine is rapidly evolving, and in the era of next-generation sequencing (NGS), in which a relatively limited amount of tissue sample can be successfully analyzed for the detection of genetic alterations, EUS TA plays a major role. Such actionable molecular changes can be seen in about one-fourth of PDAC which can be targeted with precision medicine [129]. The detection of actionable molecular alterations has led to the development of various drugs acting on one or multiple molecular pathways, like the KRAS and DNA mismatch repair (MMR) pathway, and studies have shown better overall survival when patients are treated with matched therapy (*n* = 143; 2.58 years vs. 1.51 years; hazard ratio (HR) of 0.42 (95%CI: 0.26–0.68), *p* < 0.001) [130]. Furthermore, other biomarker-based therapeutic agents are also showing promising results in clinical trials (e.g., Pembrolizumab in refractory PDAC with MSH-H/dMMR or TMB—high status) [131]. Thus, the role of adequate tissue acquisition has expanded like never before, reemphasizing the various factors affecting diagnostic adequacy. In a study by Bang JY et al., EUS FNB with a 22G Franseen needle produces a better histological core with the maintenance of tissue architecture for molecular profiling compared to an EUS FNB needle [132]. In a retrospective study, EUS FNB with a 22G Franseen needle is shown to be superior to FNA for tissue adequacy to assess microsatellite instability (88.9% vs. 35.7%, *p* =  0.03) [133]. In a recent randomized trial among 33 patients using a 22G Franseen needle with the fanning technique and stylet pull maneuver, no significant difference was found in terms of diagnostic adequacy, blood contamination, adequacy, and concentration of extracted DNA and RNA when the patients were randomized into two passes vs. three passes [134]. Recent NCCN guidelines recommend germline testing in all cases of PDAC and molecular analyses for locally advanced and metastatic cases [32]. Similarly, the American Society of Clinical Oncology (ASCO) also encourages early biomarker testing in PDAC patients who are likely to be candidates for precision-medicine-based targeted therapy after first-line chemotherapy [131]. Several other targets are underway for the development of new therapeutics. In the near future, the role of EUS FNB in precision medicine will be further explored, not only for tissue diagnosis but also for molecular profiling to improve overall survival in patients with PDAC.

### 7.2. Role of Artificial Intelligence in EUS for Pancreatic Carcinoma

Although artificial intelligence (AI) has an expanding horizon in the field of gastrointestinal endoscopy, its role in pancreatic EUS is still primitive with limited studies for the early detection of pancreatic cancer. The introduction of neural networks in AI algorithms has led to significant improvements in AI-based diagnoses of pancreatic cancer. Convolutional neural networks (CNNs) have proven to be superior over traditional neural network models like support vector machines (SVMs) due to their better discriminative function, and they are being developed for the regional recognition and classification of images for the detection of pancreatic cancer. Several studies have evaluated the role of AI in EUS for the detection of PDAC, differentiating PDAC from chronic pancreatitis (especially in complex cases in which both entities co-exist), differentiating PDAC from autoimmune pancreatitis (AIP), and the role of AI in EUS TA. For the detection of PDAC, retrospective studies have used SVMs and CAD (computer-aided design) models and have shown promising results (with a diagnostic accuracy, sensitivity, and specificity of 98%, 94.3%, and 99.5% for SVMs, and 87.5, 83.3, and 93.3% for CAD models, respectively) [135,136]. In a systematic review of 11 studies (*n* = 2292) using various AI-assisted EUS models, the overall diagnostic accuracy, sensitivity, and specificity were 80–97.5%, 83–100%, and 53–99%, respectively [137]. For differentiating PDAC from CP, several models like ANN-based digital image analyses, extended neural network (ENN)-based EUS-elastography, CAD, self-learning ANN models, and SVM predictive models have shown promising results, though most of these studies are small. In larger, multicentric prospective studies, ANN-assisted EUS-elastography models have shown an 84.3% testing accuracy and more than 90% training accuracy [138]. High values of sensitivity, specificity, and PPV were shown to differentiate PDAC from CP in another prospective study using an ANN-based model with contrast-enhanced EUS [36]. In a recent systematic review and meta-analysis, various algorithms were used for the detection of pancreatic cancer, like deep learning (DL) and artificial neural networks (ANNs). Among the five high-quality studies included in this meta-analysis, the pooled sensitivity and specificity for the AI-assisted detection of pancreatic cancer were 93% and 92%, respectively [139]. In another recent systematic review and meta-analysis of 10 studies (*n* = 1871), AI had promising results (with sensitivity, specificity, AUROC, and diagnostic OR values of 92%, 90%, 0.95, and 128.9, respectively) for the detection of pancreatic carcinoma. In this study, both ANNs and CNNs had a high diagnostic accuracy, but CNNs were not the best for the detection of pancreatic cancer [140]. More large prospective studies are needed to evaluate the role of EUS in differentiating PDAC from AIP. There are noteworthy limitations of AI-based models, like the lack of standardization for the quality of data; information bias arising due to a lack of diversity of data sources; poor-quality studies with methodological flaws; and the lack of transparent reporting. Apart from these, there are always concerns related to ethical issues and the ‘black box’ problem, in which the reasoning of the algorithm used by AI is not clear to clinicians. Despite these concerns, the use of AI as an adjunctive tool with EUS for diagnosis of PDAC, in form of detection of lesion, and AI-guided TA is likely to expand in near future. However, more large multicentric high-quality trials are needed. Until then, AI should serve as a ‘second set of eyes’ for endo-sonologists [141].

### 7.3. Role of Organoid Technology in the Diagnosis of PDAC

As PDAC is a phenotypically and genomically heterogeneous tumor, it is difficult to study the tumor behavior and treatment response ex vivo. EUS TA-derived samples have been used for developing pre-clinical models for human PDAC, which are as follows: patient-derived cells (PDCs), patient-derived tumor xenografts (PDTXs), and patient-derived organoids (PDOs) [142]. The results of PDTXs were not encouraging due to the lack of availability of a sufficient number of cells or tissue; additionally, they were more time-consuming and economically challenging. On the other hand, organoids are three-dimensional reconstructed patient-derived cancer cells that retain the genomic and transcriptomic profile of the primary tumor and can be maintained in vitro. Organoid technology has recently revolutionized the area of cancer research due to its fascinating ability to retain the genetic and phenotypic characteristics of the tumor or the primary organ, thus functionally resembling them. This makes the identification of the PDAC phenotype and therapeutic response to an anticancer drug possible in a controlled laboratory setting. Studies have demonstrated that the response to chemotherapeutic agents in organoids can successfully predict their response in patients [143,144]. This efficacy of the organoid system seems to be dependent upon the composition of the organoid [145]. In a recent feasibility study, authors were able to establish a co-culture using PDOs from a small amount of EUS FNB samples and cancer-associated fibroblasts (CAFs). Limitations like a high risk of contamination (blood as well as benign cells), compromising the yield of tissue available, the suboptimal number of needle passes, and unknown histopathological diagnoses before EUS TA can be obviated in future studies to improve the success rate [146].

### 7.4. Role of Liquid-Based Cytology

Traditionally, smears with a cytological examination, cell block test, and histopathological examination are used for sample preparation. Liquid-based cytology (LBC) has an advantage over conventional smear cytology because of several reasons, like uts higher cellularity, clearer background with fewer artifacts caused by extracellular elements like mucin and necrotic debris, and lower false negative results. Furthermore, LBC can be used for immunohistochemistry (IHC), which can be used for differentiating benign from malignant lesions, understanding tumor biology, and increasing specificity. In a meta-analysis, the EUS-guided conventional smear (CS) technique and LBC technique were compared and nine studies were included for analysis (*n* = 1308). LBC was analyzed separately based on the method used, i.e., filtration-based and precipitation-based methods. In the absence of ROSE, the precipitation-based LBC technique was found to have a higher accuracy (85.2% vs. 79.7%) and sensitivity (83.6% vs. 79.2%) compared to those of conventional techniques [116]. However, more studies are required before one can advocate LBC for routine clinical use.

### 7.5. Automated Needles for EUS TA

Most recently, an automated motorized cutting needle was devised by Limaca Medicals (Precision-GI^TM^) for more precise, quicker, and less traumatic tissue acquisition with less tissue fragmentation and blood contamination. A pilot study comparing this automated needle with a fork-tip needle has shown that the motorized needle is associated with a shorter sampling time (*p* = 0.001) and higher average histologic score (*p* = 0.002) [147].

## 8. Conclusions

EUS is currently an indispensable tool in the management of PDAC from detection and tissue diagnosis to the management of various aliments associated with PDAC. Apart from detection, with wider adoption of neo-adjuvant therapies, tissue acquisition for definitive diagnoses has become a part of routine clinical care. The tissue acquisition of histological tumor cores is likely to be a game changer due to the tremendous opportunity afforded by genomic profiling and molecular analyses to deliver precision medicine with better clinical outcomes in the near future.

## Figures and Tables

**Figure 1 diagnostics-14-00078-f001:**
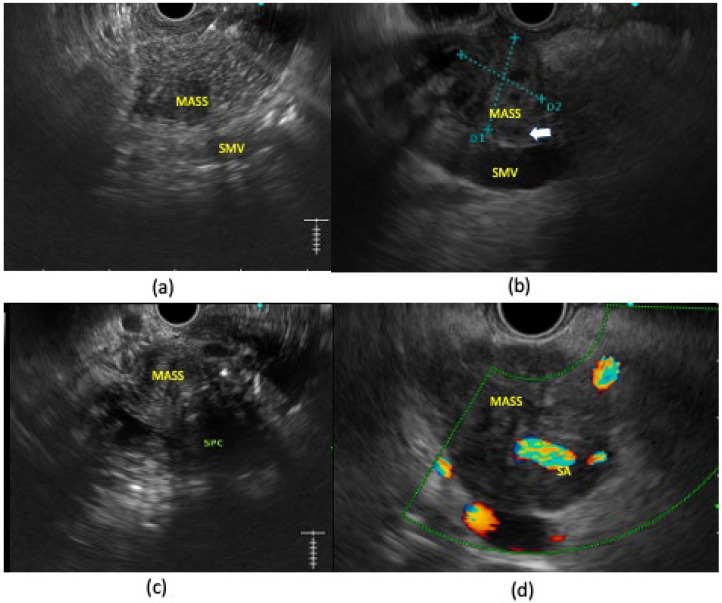
Pancreatic ductal adenocarcinoma on EUS examination with different degrees of vascular involvement. (**a**) Hypoechoic irregular mass lesion in uncinate process that has clear non-invasion with surrounding vessels. (**b**) Hypoechoic irregular mass lesion near pancreatic head that is abutting SMV; however, fat planes within the vessel are still patent (white arrow). (**c**) Heterogenous hypo-isoechoic mass lesion in pancreatic head with loss of fat planes with splenic-portal confluence. (**d**) Hypoechoic mass lesion in pancreatic body completely encasing splenic artery.

**Figure 2 diagnostics-14-00078-f002:**
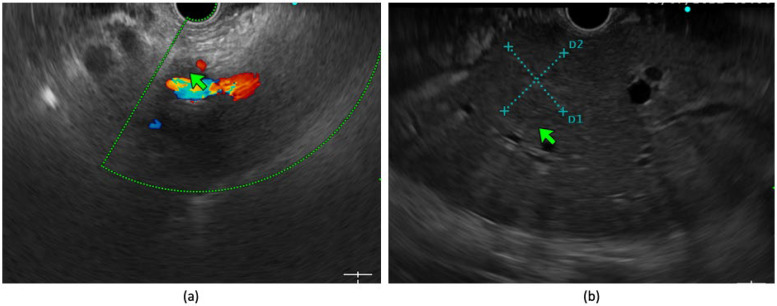
(**a**) Peri-vascular cuffing surrounding celiac axis (4.1 mm; green arrow) in a case of pancreatic adenocarcinoma. (**b**) Segment III hepatic lesion (green arrow) in a known case of pancreatic adenocarcinoma which was undetected in pre-procedure CT scan.

**Figure 3 diagnostics-14-00078-f003:**
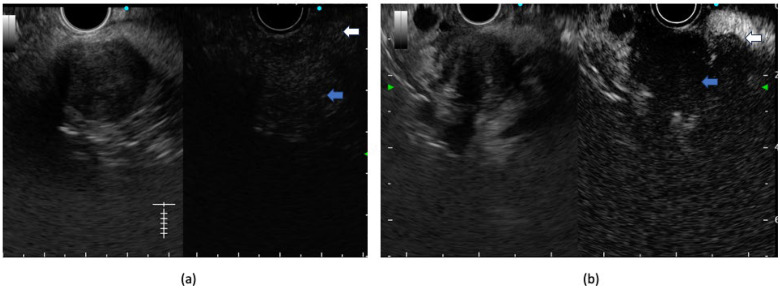
Contrast-enhanced EUS using 4.8 mL of Sonozoid showing (**a**) early and iso-enhancement of the lesion (blue arrow points towards lesion and white arrow points towards surrounding pancreatic parenchyma) in a case of pancreatic neuroendocrine tumor. (**b**) Hypo-enhancement compared to surrounding pancreatic parenchyma (blue arrow points towards lesion and white arrow points towards surrounding pancreatic parenchyma) in a case of pancreatic adenocarcinoma.

**Figure 4 diagnostics-14-00078-f004:**
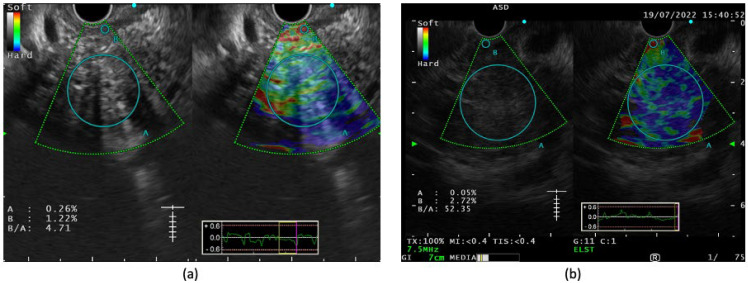
Strain elastography image showing (**a**) predominantly yellow-greenish hue (soft) with strain ratio of 4.71 in a case of chronic pancreatitis; (**b**) heterogenous blue hue (hard) with strain ratio of 52.35 in a known case of pancreatic adenocarcinoma (‘A’ is selected as the largest circumference of target lesion without involving intervening vessels and ‘B’ is selected as reference usually surrounding gastrointestinal wall).

**Figure 5 diagnostics-14-00078-f005:**
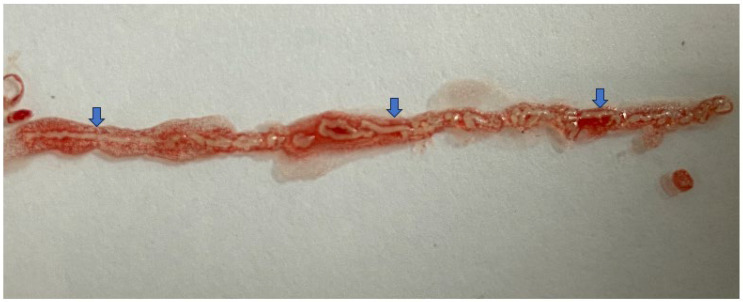
Showing macroscopic visible core (MVC) (whitish bulky tissue fragments) (blue arrow) with individual fragment >4 mm in size and total segment size of 3.2 cm (obtained using 22G Acquire needle).

**Table 1 diagnostics-14-00078-t001:** EUS tissue acquisition needles.

Type of Needle, Needle Design	Proprietary Name, Needle Diameter	Manufacturer
**EUS FNA needle**		
Menghini type	Beacon EUS Delivery system with BNX FNA preloaded needle	Beacon Endoscopic, Newton, MA, USA
	BNX FNA needle (without sheath)	Beacon Endoscopic, Newton, MA, USA
	Expect 19, 22, 25G	Boston Scientific, Marlborough, MA, USA
	Expect Flex 19G	Boston Scientific, Marlborough, MA, USA
	Expect Slimline 19, 22, 25G	Boston Scientific, Marlborough, MA, USA
	Expect Slimline Flex 19G	Boston Scientific, Marlborough, MA, USA
	ClearView 19, 22, 25G	Conmed, Billercia, MA, USA
	ClearView Sheath Stabilizer 22, 25G	Conmed, Billercia, MA, USA
	ClearView Extended Bevel 22G	Conmed, Billercia, MA, USA
	SonoTip Pro Control 19, 22, 25G	MediGlobe GmbH, Achenmühle, Germany
	EchoTip Ultra 19, 22, 25G	Cook Medical, Bloomington, IN, USA
	EchoTip Ultra coil sheath 22G	Cook Medical, Bloomington, IN, USA
	EchoTip Ultra HD Access 19G	Cook Medical, Bloomington, IN, USA
	EZ shot 2 19, 22, 25G	Olympus America, Center Valley, PA, USA
	EZ shot 2 sideport 22G	Olympus America, Center Valley, PA, USA
	EZ shot 3 plus 19, 22G	Olympus America, Center Valley, PA, USA
	EUS Sonopsy CY™ 21G	Hakko Co., Tokyo, Japan
**EUS FNB needles**		
Forward-bevel	Echotip Procore 20G	Cook Medical, Bloomington, IN, USA
Reverse-bevel	Echotip Procore 19, 22, 25G	Cook Medical, Bloomington, IN, USA
Fork-tip	SharkCore 19, 22, 25G	Medtronic, Dublin, Ireland
	Beacon EUS delivery system with SharkCore preloaded FNB needle 19, 22, 25G	Medtronic, Dublin, Ireland
Franseen	Acquire 19, 22, 25G	Boston Scientific, Marlborough, MA, USA
	Sonotip Topgain 19, 22, 25G	Mediglobe, Achenmühle, Germany

Abbreviations: EUS: Endoscopic ultrasound, FNA: Fine-needle aspiration; FNB: Fine-needle biopsy.

**Table 2 diagnostics-14-00078-t002:** Summary of recent meta-analysis of diagnostic EUS in pancreatic cancer.

SRMA (Author, year)	Number of Studies and Total Number of Patients (*n*)	Modality/Comparison	Type of Lesion	Main Outcome Measure	Sensitivity (%)	Specificity (%)	PPV	NPV	Adequacy (%)	Accuracy (%)	Contamination (%)	Other Parameters (%)	Adverse Events (%)
**Role of Diagnostic EUS**													
Rahman MIO et al., 2020 [96]	2 studies, *n* = 77	EUS vs. CECT in pancreatic protocol	Neoplastic pancreatic lesions	Diagnostic accuracy for pancreatic cancer resectability	87	63	-	-	-	-	-	Similar diagnostic OR (*p* > 0.05)	-
Krishna SG et al., 2017 [14]	4 studies	EUS after an indeterminate MDCT	Suspected pancreatic malignancies	Diagnostic performance for detection of pancreatic malignancies	85	58	77	66	-	75	-	-	-
Li Y et al., 2019 [97]	16 studies, *n* = 1325	CE EUS for pancreatic masses	Pancreatic masses	Diagnostic performance of CE EUS for the differentiation of pancreatic masses	93	84	-	-	-	-	-	LR+ 5.58 LR− 0.09 DOR 72.5%	-
Yamashita Y et al., 2019 [98]	9 studies, *n* = 887	CE EUS	Pancreatic cancer	Diagnostic performance for diagnosing pancreatic cancer	93	80	-	-	-	-	-	LR+ 4.56 LR− 0.09 DOR 59.89	-
Shin CM et al., 2023 [99]	6 studies, *n* = 430	Combined CE EUS and EUS elastography in solid pancreatic lesions	Solid pancreatic lesions	Diagnostic performance in detecting pancreatic malignancies	84	85	-	-	-	-	-	LR+ 5.31 LR− 0.15 DOR 67.72	-
Facciorusso A et al., 2021 [100]	6 studies, *n* = 701	CE EUS-guided vs. standard EUS FNA in pancreatic masses	Solid pancreatic lesions	Diagnostic outcome	84.6 vs. 75.3 (*p* < 0.001)	100% both	-	-	95.1 vs. 89.4 (*p* = 0.02)	88.8 vs. 83.6 (*p* = 0.05)	-	Histological core procurement *p* = 0.08, number of needle passes *p* = 0.29	-
**EUS tissue acquisition (EUS TA)**													
Banafea O et al., 2016 [101]	20 studies, *n* = 2761	EUS FNB	Pancreatic mass	Diagnsotic accuracy	90.8	96.5	-	-	-	91	-	LR+ 14.8 LR− 0.12 DOR 142.47	35 of 1760 patients in 15 studies
Guedes HG et al., 2018 [56]	4 studies, *n* = 504	22G versus 25G needles in EUS FNB for solid pancreatic mass	Solid pancreatic masses	Diagnostic performance	91 vs. 93 *p*>0.05	83 vs. 87 *p*>0.05	-	-	-	-	-	LR+ 4.26 vs. 4.57 LR− 0.13 vs. 0.08 *p* > 0.05	-
Xu MM et al. [102]	11 studies, *n* = 837	22G vs. 25G EUS FNA needle	Solid pancreatic lesions	Diagnostic performance	88 vs. 92 *p* = 0.046	100 vs. 100 *p* = 0.842	-	-	-	-	-	LR+ 12.61 vs. 8.44 LR− 0.16 vs. 0.13 AUSROC 0.97 vs. 0.96	-
Tian G et al., 2018 [103]	16 studies, *n* = 1824	22G vs. 25G EUS FNA needle	Masses with suspicion of pancreatic cancer	Diagnostic yield for the detection of pancreatic cancer	89 vs. 90 *p* = 0.02	100 vs. 99 *p* = 0.15	-	-	-	-	-	LR+ 485.28 vs. 59.53 LR− 0.11 vs. 0.10 AUROC 0.97 for both	-
Yang Y et al., 2016 [104]	16 studies, *n* = 828	EUS FNB	Solid malignant pancreatic lesions	Diagnostic accuracy	84	98						LR+ 8 LR− 0.17 DOR 64 AUROC 0.96	-
Reneleus et al., 2021 [63]	11 studies, *n* = 1365	EUS FNB vs. FNA	Solid pancreatic lesions	Diagnostic accuracy and safety	-	-	-	-	-	Diagnostic accuracy of 87 vs. 81 (*p* = 0.005). Cytopathological accuracy of 89 vs. 82 (*p* = 0.04). Histological accuracy of 81 vs. 74 (*p* = 0.39)	-	Mean TSR was 99% in both. Mean needle passes required for adequate tissue was 2.3 vs. 1.6 (mean difference was 0.71) (*p* < 0.0001)	2.3 vs. 1.8 (*p* = 0.64)
van Riet PA et al., 2021 [62]	18 RCTs, *n* = 2695	EUS FNB vs. FNA for sampling	Solid pancreatic and non-pancreatic lesions	Diagnostic accuracy, adequacy, number of passes, presence of tissue cores, and adverse events	-	-	-	-	90 vs. 88 (*p* = 0.76)	85 vs. 80 (*p* = 0.03) High-quality studies 82 vs. 74 (*p* = 0.002)	-	Mean number of passes was lower in FNB (mean difference −0.54) *p* = 0.03. Presence of tissue cores: 79 vs. 63 (*p* = 0.11)	0.8 vs. 1.0 (*p* = 0.8)
Hassan GM et al., 2022 [105]	9 RCTs, *n* NA	EUS FNB vs. EUS FNA	Solid pancreatic masses	Diagnostic accuracy for the diagnosis of pancreatic cancer	-	-	-	-	-	FNB had a superior accuracy compared to FNA (OR 1.87)	-	-	-
Bang JY et al., 2016 [106]	9 studies, *n* = 576	Procore vs. standard EUS FNA needle in solid lesions	All solid lesions	Diagnostic adequacy, diagnostic accuracy, acquisition of histological core tissue, and mean number of passes	-	-	-	-	75.2 vs. 89.0; OR 0.39 (*p* = 0.23)	85.8 vs. 86.2; OR 0.88 (*p* = 0.53)	-	Rate of histological core specimen acquisition (77.7% vs. 76.5%; OR 0.94, *p* = 0.85). Lower mean number of passes required for diagnosis with the ProCore needle (SMD—1.2, *p* < 0.001).	-
Li Z et al., 2022 [107]	18 studies, *n* = 2718	EUS FNB vs. EUS FNB	Pancreatic and non-pancreatic solid lesions (only solid pancreatic lesions are mentioned in the subgroup analysis)	Diagnostic accuracy, number of needle passes, adequacy, presence of tissue cores, and adverse events	-	-	-	-	FNB had a higher adequacy (RR = 0.93) *p* = 0.004	Similar pooled accuracy (RR = 0.97) *p* = 0.13	-	Fewer number of passes for adequate sampling in FNB group (MD 0.57) *p* < 0.00001. Presence of tissue core was similar (RR 0.60) *p* = 0.16	Similar (RR 1.27) *p* = 0.97
Facciorusso A et al., 2020 [108]	11 trials, 833 patients	22G FNB vs. 22G FNA needle	Solid pancreatic lesions	Diagnostic outcome and tissue adequacy	93.1 vs. 90.4	100 in both	-	-	Slightly in favour of FNB (*p* = 0.61)	-	-	No difference in histological core procurement (*p* = 0.86). Similar number of passes in FNB (MD -0.32, *p* = 0.07)	Six adverse events in FNA group and one in FNB group reported
Gkolfakis P et al., 2022 [69]	RCT 16, *n* = 1934	Different FNB needles	Solid pancreatic masses	Diagnostic accuracy (network meta-analysis)	94.6% with Franseen needle, 93.9% with Fork-tip needle, 90.4% with Menghini-tip needle, 82% with reverse-bevel needle, and 87.4% with FNA needle	Pooled specificity 100% with all needles tested	-	-	Franseen needle was better than FNA and reverse-beveled needles. Fork-tip needles were superior to reverse-beveled needle. None was superior when compared to FNA with ROSE. Both 22G and 25G Franseen needles followed by the 22G fork-tip needle showed the highest SUCRA scores concerning sample adequacy	Franseen needle was better than FNA and reverse-beveled needles. Fork-tip needles were superior to reverse-beveled needle. None was superior when compared to FNA with ROSE. The 22G Franseen needle ranked as the best FNB needle in terms of diagnostic accuracy (SUCRA score of 0.81)	-	-	Pooled rate was 2.7% with Franseen needle; 2% with Fork-tip needle; 1.3% with Menghini-tip needle; 0.8% with reverse-bevel needle, and 1.9% with the FNA needle.
Facciorusso A et al., 2019 [109]	24 studies, *n* = 6641	Franseen vs. fork-tip EUS FNB needles	Pancreatic and non-pancreatic solid lesions (only solid pancreatic lesions are mentioned in the subgroup analysis)	Sample adequacy	Similar sensitivity (95.3 vs. 93.4)	Similar specificity [100]	-	-	97 vs. 92.6 (*p* = 0.006)	96.8 vs. 95.2 (*p* = 0.8)	-	Histological core procurement was 94 vs. 93.1 (*p* = 0.7). Fewer number of passes compared to standard FNA needles (MD for Franseen was -0.44 and for Fork-tip was −1.82)	-
Facciorusso A et al., 2022 [110]	8 studies, *n* = 2147	EUS FNB with and without ROSE	Solid pancreatic lesions	Sample adequacy	94.3 vs. 91.5	-	-	-	EUS FNB with ROSE is not superior to EUS FNB alone (95.5 vs. 88.9, *p* = 0.07) especially when end-cutting needles (compared to reverse-bevel needles) are used	Superior in the EUS FNB + ROSE group (OR = 2.49, *p* = 0.03)	Number of needle passes needed to obtain diagnostic samples was not significantly different (mean difference 0.07; *p* = 0.62)	-	Only one study reported (Crino et al.)
Kong F et al., 2016 [111]	7 studies, *n* = 1299	EUS FNB with ROSE vs. EUS FNB without ROSE	Pancreatic masses	Diagnostic adequacy, yield, number of needle passes, pooled sensitivity, and specificity	91 vs. 85	100 in both	-	-	No significant difference in cytological adequacy	No significant difference in diagnostic yield	-	LR+ 28.15 vs. 29.08 LR− 0.1 vs. 0.16. Fewer needle passes in ROSE group (4 vs. 7, *p* < 0.0001)	-
Lisotti A et al., 2020 [112]	12 studies, *n* = 505	Repeat EUS FNB for the diagnosis of solid pancreatic masses	Solid pancreatic masses	Diagnostic performance of repeat EUS FNB in case of negative or inconclusive first FNA	77 (83% with ROSE)	98	99	61	-	-	-	LR+ 38.9 LR− 0.23	-
Han S et al., 2021 [113]	26 studies, *n* = 3398 (in primary NMA)	Various EUS TA needles	Solid pancreatic masses	Diagnostic accuracy compared to 22G Echotip (Cook) EUS FNA needle (NMA)	-	-	-	-	-	Performance score-wise: 22 G SharkCore FNB needle (Medtronic) > 22G EZ Shot 3 FNB needle (Olympus) > 22G Acquire FNB needle (Boston Scientific)	-	Diagnostic accuracy was not significantly different between needles with or without suction except 20G FNB needle with suction which performed significantly worse than the 22G FNA needle with suction	-
**Suction Techniques in EUS TA**													
Facciorusso A et al., 2023 [42]	9 RCTs, *n* = 756	Various EUS FNB techniques	Solid pancreatic masses	Rates of sample adequacy, blood contamination, and tissue integrity (NMA)	Modified wet suction was most sensitive (SUCRA score, 0.85) followed by slow-pull techniques and no stylet technique (SUCRA scores, 0.66 and 0.48, respectively)	-	-	-	Modified wet-suction technique was best for adequacy (SUCRA score of 0.90) followed by dry-suction and slow-pull techniques (SUCRA scores of 0.59 and 0.50, respectively)	-	Higher level of blood contamination seen with dry-suction than slow-pull technique; no-suction technique ranked as the best strategy (SUCRA score of 0.99) followed by the slow-pull technique (SUCRA score of 0.65). Modified wet-suction (SUCRA score of 0.32) and dry-suction (SUCRA score of 0.12) techniques showed poor performance in terms of blood contamination of the sample	Regarding tissue integrity, modified wet-suction technique was ranked as the best strategy (SUCRA score of 0.89) followed by slow-pull (SUCRA score of 0.66) and no-suction (SUCRA score of 0.42) techniques	Uncommon and usually mild, without significant impact on patient outcomes (abdominal pain and bleeding)
Ramai D et al., 2021 [76]	6 studies, *n* = 418	Wet vs. dry suction techniques	Solid pancreatic masses	Adequacy, sample contamination, and histological accuracy	-	-	-	-	Wet-suction technique has superior tissue adequacy (pooled adequacy rate of 91.9 vs. 77.32 (OR 3.18, *p* < 0.001))	Wet-suction technique is superior in histological diagnosis (OR of 3.68, pooled rate of 84.06 vs. 68.87, *p* < 0.001). Wet suction has superior sample quality, and accuracy	Wet-suction technique has comparable blood contamination (OR of 1.18, contamination rate of 58.33 and 54.6, *p* = 0.256)	-	-
Giri S et al., 2023 [81]	7 studies, *n* = 2048	Various suction techniques in EUS TA	Solid pancreatic and non-pancreatic lesions	Compare the diagnostic yields during EUS TA (NMA)	-	-	-	-	There was no difference between the various modalities. For the SUCRA analysis, WS > SSP > DS > NS	No significant difference in ORs of adequacy when adjusted for either of the needle types. For the SUCRA analysis, WS > NS > DS > SSP	When adjusting for FNA needle, there was no difference between the interventions	No significant difference between the studies with respect to moderate-to-high cellularity of samples	-
**EUS TA in Presence of Biliary Stents**													
Facciorusso A et al., 2023 [114]	7 studies, *n* = 2458	EUS TA in presence and absence of biliary stent	Solid pancreatic head masses	Diagnostic accuracy before and after biliary stenting in jaundiced patients with pancreatic head masses	Overall diagnostic sensitivity lower in biliary stent group (82.9 vs. 87.5; OR 0.59; *p* < 0.001); in SEMS subgroup (*p* = 0.006) but not in plastic stent group (*p* = 0.12)	-	-	-	No significant difference in adequacy (*p* = 0.81)	No overall significant difference 85.4 vs. 88.1 (*p* = 0.07). No significant difference in plastic stent vs. no stent (*p* = 0.67). Significant difference in SEMS vs. no SEMS (*p* = 0.05)	-	No significant difference in number of needle passes (*p* = 0.38)	No significant difference (*p* = 0.75)
Giri S et al., 2023 [115]	9 studies, *n* = 3257	EUS TA in presence and absence of biliary stent	Pancreatic masses undergoing EUS TA	Diagnostic accuracy of EUS TA in presence and absence of biliary stent	79 vs. 88; Using non-strict criteria in patients with stents, the sensitivity was lower with metal stents than with plastic stents (83% vs. 90%)	-	-	-	Comparable in stent vs. non-stent groups and in plastic and SEMS group	Lower accuracy with stent (OR of 0.58) using non-strict criteria and comparable sensitivity between metal stents and plastic stents	-	Patients with stents required greater number of passes (MD = 0.31)	-
**Advances of EUS**													
Chandan S et al., 2020 [116]	9 studies, *n* = 1308	EUS-guided precipitation-based LBC conventional smear	Solid pancreatic masses	Diagnostic yield of EUS-guided conventional smear vs. LBC	Precipitation based LBC higher sensitivity (85.2 vs. 79.7)	Precipitation based LBC comparable specificity (99.5 vs. 99.4)	Precipitation based LBC comparable PPV (99.5 in both)	NPV was found to be higher with filtration-based LBC technique (50.9%) as compared with CS (46.2%) and precipitation-based LBC techniques (35.4%).	-	Precipitation-based LBC had a higher accuracy	-	-	-
Prasoppokakorn T et al., 2021 [117]	8 studies, *n* = 870	AI-assisted diagnosis of PDAC by EUS	Pancreatic mass	AI-assisted B-mode EUS sensitivity and specificity 90%, 91% respectively. AI-assisted CE EUS sensitivity and specificity 95%, 95% respectively. AI-assisted EUS elastography sensitivity and specificity 88%, 83% respectively.	AI-assisted EUS 91% AI-assisted B-mode EUS 91%	AI-assisted EUS 90% AI-assisted B-mode EUS 90%	AI-assisted B-mode EUS 94%	AI-assisted B-mode EUS 84%	-	-	-	-	-
Dhali A et al., 2023 [118]	21 studies	AI-assisted vs. conventional EUS for detection of pancreatic SoLs	Diagnostic performance	Higher accuracy of AI-assisted EUS for detection and differentiation	93.9	93.1	91.6	93.6	-	93.6	-	-	-

Abbreviations: PDAC—Pancreatic ductal adenocarcinoma; EUS—Endoscopic ultrasound; CE EUS—Contrast-enhanced EUS; EUS TA—EUS-guided tissue acquisition; FNA—Fine-needle aspiration; FNB—Fine-needle biopsy; LBC—Liquid-based cytology; CS—Conventional smear; ROSE—Rapid on-site cytology evaluation; MOSE—Macroscopic on-site cytology evaluation; SEMS—Self-expanding metal stent; WS—Wet-suction method; SSP—Stylate slow-pull method; DS—Dry-suction method; AI: Artificial intelligence; PPV—Positive predictive value; NPV—Negative predictive value; OR—Odd’s ratio; RR—Relative risk; AUROC—Area under receiver operating curve; NMA—Network meta-analysis; LR—Likelihood ratio.

## Data Availability

Not applicable.

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
