# Peer review of "Role of Endoscopic Ultrasound in Diagnosis of Pancreatic Ductal Adenocarcinoma"

_diagnostics, 2023, doi:10.3390/diagnostics14010078_

Round 1
Reviewer 1 Report (Previous Reviewer 1)
Comments and Suggestions for Authors
The authors have addressed all of my concerns.
Author Response
Ans: Thank you for the kind comments.
Reviewer 2 Report (Previous Reviewer 2)
Comments and Suggestions for Authors
The authors took into account my previous recommendations and modified the manuscript accordingly.
Comments on the Quality of English LanguageMinor editing is required.
Author Response
The authors took into account my previous recommendations and modified the manuscript accordingly
Ans: Thank you for the kind comments.
Reviewer 3 Report (New Reviewer)
Comments and Suggestions for Authors
This is a comprehensive review of the role of EUS in pancreatic duct adenocarcinoma. It is a very good effort, cover all aspects of the role of EUS in diagnosis of PDAC. I just have few comments"
Few language mistakes:
Line 179: CH-EUS----------> EC-EUS.
Line 226: routine------------> to be removed or corrected.
Line 570: (LBC)--------------> This abbreviation is previously mentioned in the same paragraph in line 564. So, remove this abbreviation in line 570.
I think not everyone is oriented with EUS and its new advantages, even those who have some experience in the technique. I suggest that the authors add a paragraph before the conclusion to summarize what is going on in his place, what needle he is using, what technique of suction he prefers, is he uses ROSE,... This may summarize a lot of date for the readers.
Comments on the Quality of English Language
OK.
Author Response
This is a comprehensive review of the role of EUS in pancreatic duct adenocarcinoma. It is a very good effort, cover all aspects of the role of EUS in diagnosis of PDAC. I just have few comments"
Ans: Thank you for the kind comments.
Few language mistakes:
Line 179: CH-EUS----------> EC-EUS.
Line 226: routine------------> to be removed or corrected.
Line 570: (LBC)--------------> This abbreviation is previously mentioned in the same paragraph in line 564. So, remove this abbreviation in line 570.
Ans: Thank you for pointing out these errors. We have corrected the same in the revised manuscript.
I think not everyone is oriented with EUS and its new advantages, even those who have some experience in the technique. I suggest that the authors add a paragraph before the conclusion to summarize what is going on in his place, what needle he is using, what technique of suction he prefers, is he uses ROSE,... This may summarize a lot of date for the readers.
Ans: We have added one paragraph on personal comments in the revised manuscript.
This manuscript is a resubmission of an earlier submission. The following is a list of the peer review reports and author responses from that submission.
Round 1
Reviewer 1 Report
Comments and Suggestions for Authors
Chatterjee and Shah reviewed on the role of endoscopic ultrasound in the diagnosis of pancreatic ductal adenocarcinoma. The authors covered the recent findings in the detection and the staging of pancreatic tumor, the contrast enhanced EUS, EUS elastography, and EUS guided tissue acquisition. The authors highlighted the potential role of tissue acquisition using EUS. This is a current topic, of interest for physicians, was comprehensively designed and conducted. The contents presented here is of important practical values and inspiring for physician investigators. This reviewer has some minor suggestions.
1, In the introduction, the authors may discuss the current method for the diagnosis and prognosis of PDAC: Xu X, Liang JH, Li JH, Xu QC, Yin XY. Values of a novel pyroptosis-related genetic signature in predicting outcome and immune status of pancreatic ductal adenocarcinoma. Gastroenterol Rep (Oxf). 2022 Sep 29;10:goac051. doi: 10.1093/gastro/goac051. PMID: 36196256; PMCID: PMC9522386.
2, Figure 1, the panels can be re-arranged as 2 by 2.
3, All figures: it is weird that the labels of the panels are placed below the graphs, and the figure legends on top of the graphs.
4, EUS-TA is not defined in the main text.
5, EUS-TA could facilitate the preparation of organoids for diagnosis and research. Suggest the authors to briefly discuss this point: Chen S, Wang M, Liu L, Wang G, Wang L, Zhong C, Gao C, Wu W, Li L. Ultrasound-guided fine-needle aspiration/biopsy-based pancreatic organoids establishment: an alternative model for basic and preclinical research. Gastroenterol Rep (Oxf). 2023 Apr 10;11:goad019. doi: 10.1093/gastro/goad019. PMID: 37051577; PMCID: PMC10085542.
Author Response
Reviewer 1:
Chatterjee and Shah reviewed on the role of endoscopic ultrasound in the diagnosis of pancreatic ductal adenocarcinoma. The authors covered the recent findings in the detection and the staging of pancreatic tumor, the contrast enhanced EUS, EUS elastography, and EUS guided tissue acquisition. The authors highlighted the potential role of tissue acquisition using EUS. This is a current topic, of interest for physicians, was comprehensively designed and conducted. The contents presented here is of important practical values and inspiring for physician investigators. This reviewer has some minor suggestions.
Ans: Thank you for the kind comments and providing us valuable inputs.
1, In the introduction, the authors may discuss the current method for the diagnosis and prognosis of PDAC: Xu X, Liang JH, Li JH, Xu QC, Yin XY. Values of a novel pyroptosis-related genetic signature in predicting outcome and immune status of pancreatic ductal adenocarcinoma. Gastroenterol Rep (Oxf). 2022 Sep 29;10:goac051. doi: 10.1093/gastro/goac051. PMID: 36196256; PMCID: PMC9522386.
Ans: Though it’s an important study, genetic signature in pancreatic carcinoma is very vast and different topic all- together which is beyond the scope of this review article on utility of EUS in diagnosis of pancreatic carcinoma, thus we have not included the same in the current article.
2, Figure 1, the panels can be re-arranged as 2 by 2.
Ans: We have re-arranged the panel as per suggestions in the revised manuscript.
3, All figures: it is weird that the labels of the panels are placed below the graphs, and the figure legends on top of the graphs.
Ans: We have re-arranged figures legends below the figure in the revised manuscript.
4, EUS-TA is not defined in the main text.
Ans: We have defined EUS-TA in the revised manuscript.
5, EUS-TA could facilitate the preparation of organoids for diagnosis and research. Suggest the authors to briefly discuss this point: Chen S, Wang M, Liu L, Wang G, Wang L, Zhong C, Gao C, Wu W, Li L. Ultrasound-guided fine-needle aspiration/biopsy-based pancreatic organoids establishment: an alternative model for basic and preclinical research. Gastroenterol Rep (Oxf). 2023 Apr 10;11:goad019. doi: 10.1093/gastro/goad019. PMID: 37051577; PMCID: PMC10085542.
Ans: Thank you for pointing out this important lacune. We have added one separate section on Organoids (Section 7.3) and incorporated above mentioned studies in the revised manuscript.
Reviewer 2 Report
Comments and Suggestions for Authors
The narrative review provided is a comprehensive statement of the use of EUS in pancreatic adenocarcinoma. While some chapters of the current review are up-to-date and provided sufficient information others are very thin described.
I would recommend that chapter 7 Recent advancement in EUS-TA in pancreatic carcinoma should contain some additional information.
- In the „role of EUS-FNA in the era of precision medicision” I also recommend including and commenting on the use of organoids based on EUS-FNB tissue harvesting to from PDAC. PMCID: PMC10377599 DOI: 10.3390/cancers15143677
- In the Role of AI in EUS pancreatic carcinoma the authors should discuss more the ability to distinguish between CP and PDAC on EUS imaging PMCID: PMC8870917 DOI: 10.3390/diagnostics12020309
Minor corrections are required
Author Response
Reviewer 2:
- In the „role of EUS-FNA in the era of precision medicision” I also recommend including and commenting on the use of organoids based on EUS-FNB tissue harvesting to from PDAC. PMCID: PMC10377599DOI: 3390/cancers15143677
Ans: Thank you for pointing out this important lacune. We have added one separate section on Organoids (Section 7.3) and incorporated above mentioned studies in the revised manuscript.
- In the Role of AI in EUS pancreatic carcinoma the authors should discuss more the ability to distinguish between CP and PDAC on EUS imaging PMCID: PMC8870917DOI: 3390/diagnostics12020309
Ans: We have incorporated above mentioned studies in the revised manuscript.
Reviewer 3 Report
Comments and Suggestions for Authors
Comments and Suggestions for Authors
The study by Chatterjee et al. on the Role of Endoscopic Ultrasound in the diagnosis of Pancreatic Adenocarcinoma is an important issue. There are some flaws in their study that should be improved before the publication of the study. This study lacks novelty, and I address the remaining concerns regarding this study.
1. The authors need to check the inconsistency in the references on line 36.
2. The introduction is short and requires additional information. DOI: 10.3389/fnut.2022.1078642
3. Correct the sentence in line # 57.
4. Elaborate the first-time using abbreviations such as MRI and CT scans in the text.
5. Correct the statement in line #70-71.
6. Remove inconsistency in spacing in line # 138 and remove it from the entire manuscript.
7. Major English corrections are required by native speaker or professionals.
8. What is the novelty and significance of this study? Little mechanistic information is available in the whole manuscript. After reviewing the entire article, it seems that it is not suitable for publication.
Comments on the Quality of English LanguageMajor English corrections are required by native speaker or professionals.
Author Response
Reviewer 3:
- The authors need to check the inconsistency in the references on line 36.
Ans: Thank you for pointing this error. We have corrected the same in the revised manuscript.
- The introduction is short and requires additional information. DOI: 10.3389/fnut.2022.1078642
Ans: Thank you for providing this inputs. We have expanded the introduction in the revised manuscript. However, regarding suggested article, genetic signature in pancreatic carcinoma is very vast and different topic all- together which is beyond the scope of this review article, thus we have not included the same in the current article.
- Correct the sentence in line # 57.
Ans: We have corrected the line no 57 in the revised manuscript.
- Elaborate the first-time using abbreviations such as MRI and CT scans in the text.
Ans: We have provided full form of all abbreviations during its first time use in the revised manuscript.
- Correct the statement in line #70-71.
Ans: We have corrected the line no 70-71 in the revised manuscript.
- Remove inconsistency in spacing in line # 138 and remove it from the entire manuscript.
Ans: We have done uniform spacing throughout the revised manuscript.
- Major English corrections are required by native speaker or professionals.
Ans: We have done Major English correction by a professional in the revised manuscript. Thank you for providing us feedback to improve our manuscript.
- What is the novelty and significance of this study?
Ans: As it is not an original manuscript, it contains only data which have been published previously. The objective of this review article was to provide up-to date information available on latest advances on role of EUS in diagnosis of PDAC which we have done extensively and diligently.